# Multisensory-motor integration in olfactory navigation of silkmoth, *Bombyx mori*, using virtual reality system

**Mayu Yamada**[1†], **Hirono Ohashi**[1†], **Koh Hosoda**[1‡], **Daisuke Kurabayashi**[2‡], **Shunsuke Shigaki**[1*†]

[1]Graduate School of Engineering Science, Osaka University, Osaka, Japan;
[2]Department of Systems and Control Engineering, Tokyo Institute of Technology, Tokyo, Japan

**Abstract** Most animals survive and thrive due to navigational behavior to reach their destinations. In order to navigate, it is important for animals to integrate information obtained from multisensory inputs and use that information to modulate their behavior. In this study, by using a virtual reality (VR) system for an insect, we investigated how the adult silkmoth integrates visual and wind direction information during female search behavior (olfactory behavior). According to the behavioral experiments using a VR system, the silkmoth had the highest navigational success rate when odor, vision, and wind information were correctly provided. However, the success rate of the search was reduced if the wind direction information provided was different from the direction actually detected. This indicates that it is important to acquire not only odor information but also wind direction information correctly. When the wind is received from the same direction as the odor, the silkmoth takes positive behavior; if the odor is detected but the wind direction is not in the same direction as the odor, the silkmoth behaves more carefully. This corresponds to a modulation of behavior according to the degree of complexity (turbulence) of the environment. We mathematically modeled the modulation of behavior using multisensory information and evaluated it using simulations. The mathematical model not only succeeded in reproducing the actual silkmoth search behavior but also improved the search success relative to the conventional odor-source search algorithm.

**\*For correspondence:**
shigaki@arl.sys.es.osaka-u.ac.jp

†These authors contributed equally to this work
‡These authors also contributed equally to this work

**Competing interest:** The authors declare that no competing interests exist.

## Editor's evaluation

This paper uses a multi-model virtual reality system to assess which combinations of visual, wind, and olfactory information male silk moths rely on to find a female. The overall conclusion is that for the moths to search effectively, wind direction information is an important input. Vision, on the other hand, while it is used to control angular velocity, does not appear to be important for the moths to search effectively. This paper is of interest to neuroscientists and engineers interested in how multimodal sensory input controls navigational behavior. The experiments and modeling effort provide an advance in our understanding of how odor and wind information is combined in male silkmoths as they search for females.

## Introduction

In many organisms, including humans, appropriate behavior is determined based on the integration of different kinds of information from the environment. Examples of information obtained from the environment include light, sound, odor, and wind. Unlike physical signals that are transmitted as waves,

it is difficult to obtain precise directional and spatial information for odors because they (chemical substances) are carried by the wind. However, odors are widely utilized as a communication tool by organisms (*Renou, 2014*) because they have good residuality and diffusivity that physical wave signals do not have. Insects, in particular, communicate extensively using odor (e.g. aggregation pheromones, trail pheromones, sex pheromones; *Wyatt, 2014*), despite their small-scale neural systems. Odor information is also largely used to locate feeding sites and flowers (*Renou, 2014*).

Understanding odor-based search behavior is of great value not only in biology, but also in engineering research. This is because odor-based search can be applied to the use of gas leak source search robots or lifesaving robots in a disaster area instead of dogs. Such odor-based search behavior has been reported to be observed in walking and flying insects such as cockroaches, beetles, moths, and flies, and detailed behavioral analyses have also been conducted (*Willis et al., 2011*; *Burkhardt et al., 1967*; *Ryohei et al., 1992*; *Willis and Arbas, 1991*; *Saxena et al., 2018*; *Pang et al., 2018*). For example, flying moths (*Manduca sexta L.*) or cockroaches (*Periplaneta americana*) move upwind when they encounter an odor, and move in a crosswind direction when they lose the odor. Moreover, walking moths (*Bombyx mori*) and dung beetles (stercorarius) perform a 'recapture' or 'reenter' of the odor plume by combining straight and zigzag movements. They have excellent performance in searching for feeding grounds, nests, and mating partners using this olfactory behavior (*Reisenman et al., 2016*). Although there have been many attempts to model the excellent odor-source search behavior of insects and implement them in robots (*Chen and Huang, 2019*), artificial systems have not yet achieved the same abilities as insects. One of the reasons for this lack of performance is that models do not incorporate when and under what conditions and which sensory information is fed back to inform subsequent behavior. For example, in order to move into an environment with high uncertainty, an agent needs to modulate its speed of movement according to the type and amount of sensory input, but many models have insufficient discussion on speed modulation. To solve this problem, it is necessary to measure behavioral changes in insects when multiple types of sensory information are presented to them. Previously, *Pansopha et al., 2014* found that the mating behavior of a silkmoth (elicited by odor stimuli) was modulated by visual stimuli. Further, in a study on crickets, *Haberkern and Hedwig, 2016* reported that long-term tactile stimulation suppressed phototaxis, suggesting that behavioral switching between proximal environmental information and phototaxis may occur. *Duistermars and Frye, 2008* showed that by providing visual stimuli using optical flows to *Drosophila* in addition to odor stimuli, spatial information, such as the orientation of the odor source, could be obtained. These results reveal that behavioral modulation and switching mechanisms occur in insects through the acquisition of multiple types of environmental information. Because these experimental results were obtained when controlled stimuli were provided, the mechanisms of behavioral modulation and switching when complex environmental changes are presented are still unknown. Because the search behavior of insects is expected to depend on the environmental dynamics of the search, there should be a difference between search behavior in a relatively well-organized airflow environment and a turbulent environment. However, it is difficult to obtain the relationship between the sensory input and behavioral output of insects moving in a turbulent environment (*Baker et al., 2018*).

In recent years, the use of virtual reality (VR) systems in insect behavior experiments has attracted attention as a way of presenting complex environmental changes. Generally speaking, VR is a system to achieve (or measure) a purposeful behavior by connecting a device that can provide multisensory stimuli to the organisms and a space in which the organisms can move virtually (*Zhou and Deng, 2009*). Based on the general definition of VR, previous experiments in which stimuli were presented to multi-sensory organs of an organism did not provide a space for the organism to virtually perform the task. Therefore, even if we could measure the behavioral changes when stimuli were input to multi-sensory organs, it would be difficult to directly discuss which part of the task the behavioral changes were contributing to. By converting the behavioral experiment of the organisms into VR, we expect to clarify how the behavioral modulation mechanism using multisensory information contributes to the function of navigation. VR has therefore been proposed to investigate the navigation of mammals and invertebrates (e.g., *Naik et al., 2020*; *Radvansky and Dombeck, 2018*). For example, *Kaushik et al., 2020* found, using an insect VR system, that dipterans use airflow and odor information for visual navigation. Moreover, *Radvansky and Dombeck, 2018* succeeded in measuring the olfactory navigation behavior of mammals (mice) using a VR system. All motile organisms use spatially distributed

chemical features of their surroundings to guide their behavior; however, investigating the principles of this behavioral elicitation has been difficult because of the technical challenges in controlling chemical concentrations in space and time during behavioral experiments. Moreover, their research has demonstrated that the introduction of VR into the olfactory navigation of organisms can solve the above problem. Thus, VR systems allow for a more natural presentation of environmental changes, as well as a quantitative analysis of the effects of motion modulation and switching mechanisms on functions such as search and navigation. Furthermore, they allow researchers to create and test insects in situations that do not occur in nature and may therefore play an important role in the construction of robust behavioral decision algorithms for unknown environments.

In order to elucidate the adaptive odor-source search behavior of insects, we constructed a VR system that can present multiple types of environmental information simultaneously and continuously, and we employed it to clarify how sensory information other than odor is used. Our VR system was connected to a virtual field built in silico and presented odor, wind, and visual stimuli according to environmental changes in the virtual search field. We employed an adult male silkmoth (*Bombyx mori*) as our measurement target. Female search behavior of the silkmoth has a stereotypic pattern (*Ryohei et al., 1992*), but the duration and speed of the behavior are regulated by the frequency of odor detection and the input of other sensory information (*Pansopha et al., 2014*; *Shigaki et al., 2019*; *Shigaki et al., 2020*). However, previous studies have observed silkmoth behavior in response to a specified, unchanging amount of the stimulus, and the nature of behavioral changes during an actual odor source search warrants further investigation.

In this study, we used a VR system to clarify which sensory organs the silkworm moth uses to search for females and how they are used. In addition, we constructed a model from our biological data and tested the validity of the model using a constructive approach.

## Results

In this study, we analyzed changes in the behavior of an adult male silkmoth in his search for females in response to multiple sensory inputs. To measure behavior, we constructed a novel virtual reality device that presents odor, visual, and wind stimuli (*Figure 1A*) to provide the silkmoth with the illusion that it was searching for a female (see *Figure 1—video 1*). We provided the silkmoth with odor, wind, and visual stimuli (*Figure 1B*). In order to provide the odor stimulus to the left and right antennae independently, we designed two odor discharge ports above each antenna. We integrated these discharge ports with a tethered rod that was fixed to the silkmoth body. We provided the wind stimulus to the silkmoth from four directions: front, back, left, and right. Moreover, the visual stimulus provided an optical flow in the direction opposite to the turn direction of the silkmoth. In this experiment, we focused on odor-based navigation instead of visual object recognition, so that we adopted an optical flow to show which direction the landscape was flowing. The VR device was connected to a virtual field built in silico, and the odor puffs in the virtual field were produced by image processing of the actual odor diffusion using an airflow visualization system (*Yanagawa et al., 2018*; *Connor et al., 2018*). In addition, the virtual field was set with wind flowing uniformly from left to right. The search performance of the silkmoth using the constructed VR system was the same as that in free-walking experiments (see Supplementary Materials). We carried out experiments using the VR by changing the number of sensory inputs with (1) two or less types of sensory input (Group 1: odor only/odor and wind/odor vision), and (2) three types of sensory input (Group2: odor wind vision) (*Figure 1C*). Three repetitions of the experiment were conducted using 10 silkworm moths for each environmental condition ($n = 30$). We set a time limit of 300 s because theoretically, the silkmoth could reach the odor source in an infinite time. The search was considered a failure if the moth did not enter a radius of 10 mm from the odor source within the time limit.

### Wind and visual effects on the olfactory navigation

We compared the search performance in response to different types of stimuli by presenting wind and visual stimuli in addition to olfactory stimuli in all conditions. We first measured the behavior when wind or visual stimuli were provided in addition to an odor stimulus (Group 1). The blue and black lines in *Figure 2B—F* are the trajectories of successful and unsuccessful searches, respectively. The amplitude of the heading angle histogram (*Figure 2B—F*; displayed in polar coordinates) indicates the

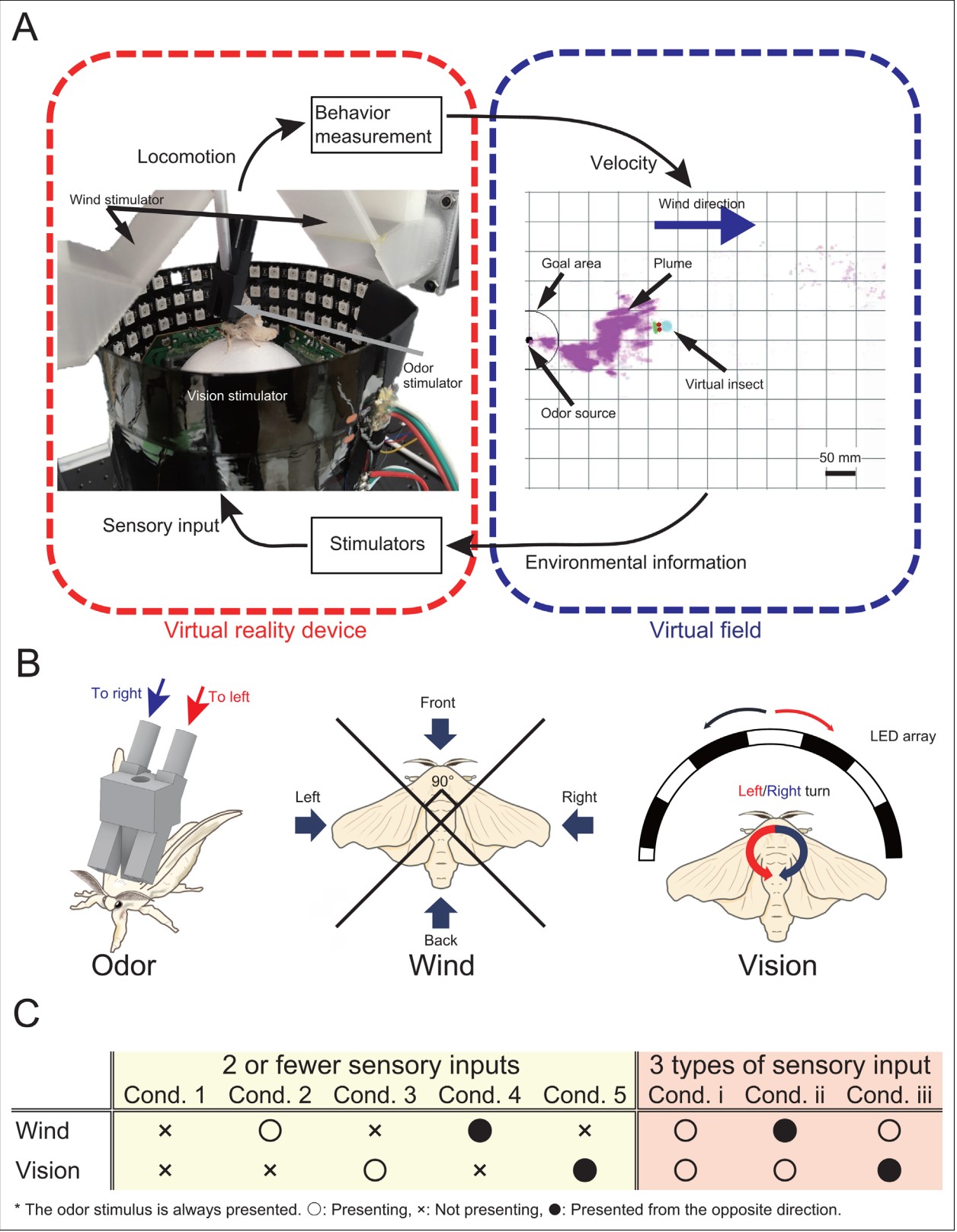

**Figure 1.** The virtual reality (VR) system for olfactory navigation of the insect and a list of experimental conditions. (**A**) The VR system is equipped with a stimulator of odor, vision, and wind, and is connected to a virtual odor field. The insect on the VR device performs olfactory navigation in a virtual space. (**B**) Definition of the way of presentation of each sensory stimulus. (**C**) The odor is presented under all conditions. 'O', '×', and '●' indicate presented, not presented, and presented from a direction opposite to the actual direction, respectively.

*Figure 1 continued on next page*

*Figure 1 continued*

The online version of this article includes the following video and figure supplement(s) for figure 1:

**Figure supplement 1.** System configuration diagram and evaluation of virtual reality (VR) system.

**Figure 1—video 1.** A video of a silkmoth behavior experiment using a virtual reality (VR) system.

https://elifesciences.org/articles/72001/figures#fig1video1

**Figure supplement 2.** Plume capturing method and odor frequency property of plume.

frequency. Judging from the trajectory and heading angle histogram, the trajectories in cond. 1 (only odor) show that although the silkmoth moved toward the upwind direction with a high frequency, it deviated from the range where the odor had a high probability of reaching (high odor area), and the search failed (*Figure 2B*). By presenting wind (cond. 2) and visual (cond. 3) information in addition to odor, the silkmoth reduced the deviations from the high odor area, which qualitatively indicates that the odor-source search behavior was modulated by other sensory information (*Figure 2CD*). In the case of wind input, even if there was a deviation, modulation was elicited to change the heading

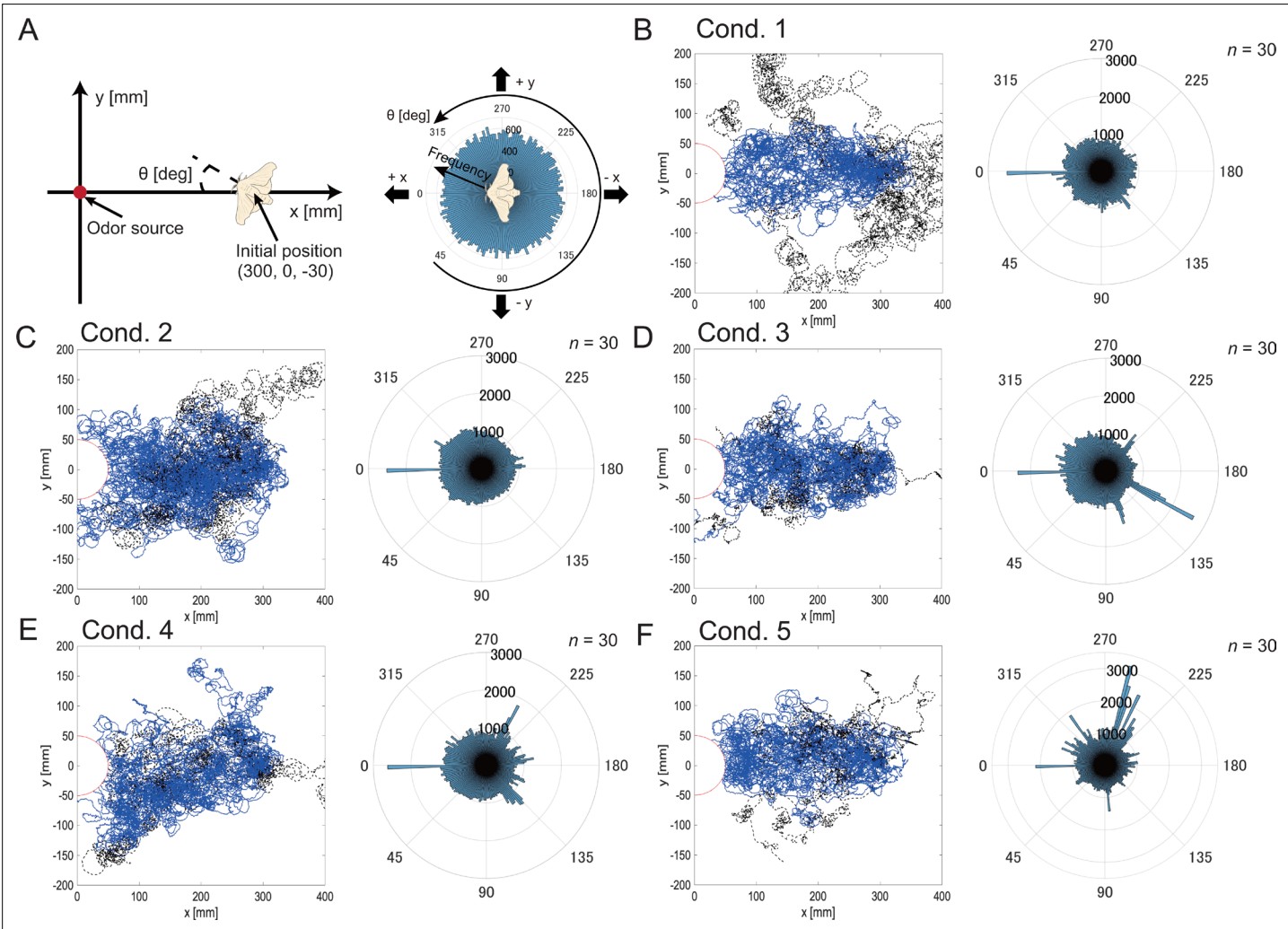

**Figure 2.** Definition of odor-source search experimental fields and initial heading angle (**A**), and the result of trajectories and heading angle histograms under each experimental condition (**B—F**). The blue and black lines are the trajectories of successful and unsuccessful searches, respectively. Moreover, the amplitude of the heading angle histogram displayed in polar coordinates indicates the frequency. The higher the frequency, the more he silkmoth moved in that direction. Note that the 0°direction is the windward direction and does not necessarily move toward the odor source.

The online version of this article includes the following figure supplement(s) for figure 2:

**Figure supplement 1.** Odor-source search performance of a silkmoth when the direction of odor stimulation is lost.

angle to the upwind direction. In the case of visual input, although there was no trajectory deviating away from the high-odor area, because the frequency of the heading angle histogram was also high at angles other than $0°$, the posture control makes the search behavior more careful. According to the heading angle histogram of cond. 4, which was presented from the direction opposite to the direction in which the wind actually blew, there was also a peak in the downwind direction; This indicates that the behavioral modulation was affected by the wind information (*Figure 2E*). Even when the visual stimulus direction was reversed, the peak in the upwind direction was not extreme, and a peak was generated in the crosswind direction (*Figure 2F*). By presenting the visual information in the opposite direction (the visual information was input as if it were rotating in the opposite direction), the illusion is that it was not rotating correctly, and more rotational behavior was induced. In order to quantitatively evaluate how similar these search trajectories were for each condition, we performed a two-dimensional histogramization of the trajectories and a calculation of similarity.

We created a migration probability map in order to visualize the effects of differences in environmental conditions on the behavioral trajectory (*Figure 3A—E*). To quantitatively evaluate the similarity between the migration probability maps, we calculated the earth mover's distance (EMD) (*Rubner et al., 1997*). EMD is an index that calculates whether the distributions of two histograms are similar. The smaller the EMD value, the more similar the two histograms; the larger the value, the less similar they are. *Figure 4F* shows the result of calculating EMD based on cond. 1, in which only the odor stimulus was presented. The silkmoths' trajectories in conds. 3 and 5 (odor + vision) were very similar to the trajectory in cond. 1 (only odor), suggesting that vision did not significantly affect search behavior. In addition, the EMD value when wind stimulus was presented in addition to odor was 10, suggesting that the addition of wind stimulus significantly affected search behavior. We illustrated the navigation success rate and search time (yellow background in *Figure 4AB*). Additionally, we calculated the search success rate per unit time (SPT) as a measure of the relationship between the search success rate and search time (yellow background in *Figure 4C*). Higher SPT values represent better search performance. The success rate was most improved when the wind was presented correctly (cond. 2) compared to when only the odor stimulus was provided (cond. 1). The success rate was lower under the condition where the wind was presented from the direction opposite to the direction detected in the virtual environment (cond. 4), compared with the condition where only the odor stimulus was presented (cond. 1). Moreover, SPT in conds 3 and 5 (in which the direction of the visual stimulus was changed without presenting the wind stimulus) was almost the same (0.411 vs 0.409), suggesting that the visual stimulus had little effect on search performance. Therefore, the success rate of navigation tends to improve by correctly presenting a wind stimulus in addition to the odor.

We found that when three types of sensory stimuli were presented (Group 2), the success rate changed significantly compared to when two types of sensory stimuli were presented (red background in *Figure 4*). Focusing on cond. i, the SPT values were also significantly different, suggesting that olfactory navigation can be performed more accurately and efficiently by presenting all sensory stimuli. Additionally, under the condition that the wind stimulus is presented from the direction opposite to the direction received in the virtual environment, the search success rate is significantly reduced, and it is clear that information on wind direction is an important factor. We therefore hypothesized that the silkmoth performs efficient female searches using all sensory inputs: odor, wind, and vision.

## Extraction of behavioral modulation mechanisms in the odor-source search

In the previous section, it was found that information on wind direction, in addition to odor, contributed to improving the success rate in searching for the odor source. Here, we analyze in detail how behavior is modulated by visual and wind information using three experimental conditions: a forward condition (cond. i), an inverse condition for wind direction information (cond. ii), and an inverse condition for visual stimuli (cond. iii).

We first analyzed the effect of information on wind direction on odor-source search behavior. Specifically, we analyzed whether the movement speed changed between cond. i and cond. ii because the wind causes behavioral modulation. We calculated the movement speed change with respect to the odor detection frequency (Hz) because it has been reported that the odor detection frequency is related to the distance from the odor source (*Kikas et al., 2001*; *Figure 5AB*). Here, odor detection frequency is defined as the frequency at which the silkmoth receives odors per unit of time. The odor

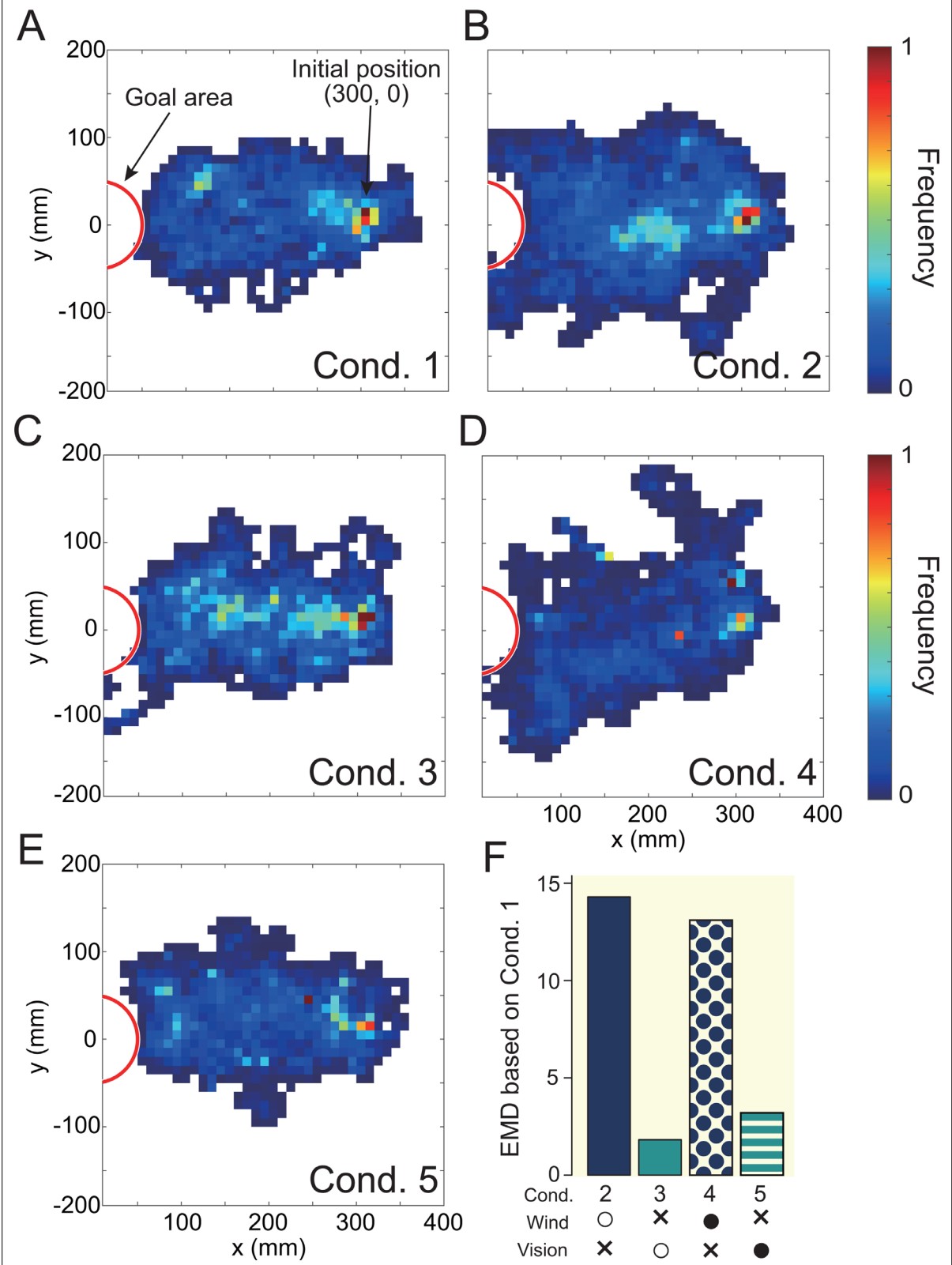

**Figure 3.** The result of visually expressing the trajectory under each experimental condition with a migration probability map (**A—E**). The white area in the figure indicates that the moth did not move into that space. A quantitative evaluation using earth mover's distance (EMD) is illustrated in **F**, which is the result of calculating the similarity (EMD) based on the trajectory of cond. 1 (odor presentation only). The lower the value, the higher the similarity, and the higher the value, the lower the similarity of the trajectories.

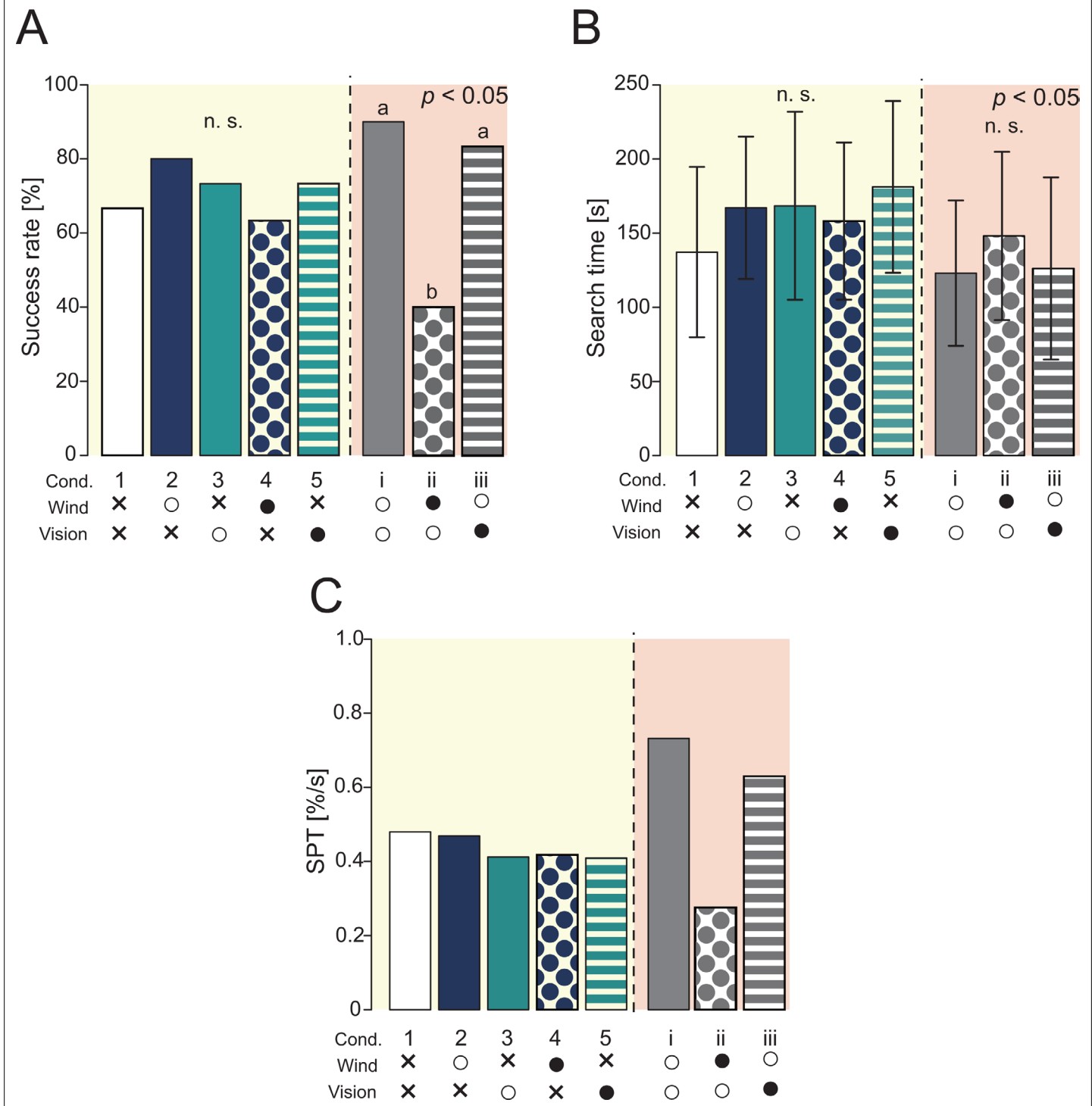

**Figure 4.** Results of navigation experiments using virtual reality (VR). The yellow background is the result of providing two or fewer sensory stimuli, and the red background is the result of providing three sensory stimuli. (**A**) The success rate of the navigation (Fisher's exact test, p < 0.05). (**B**) The search time at the time of success (Steel-Dwass test, p < 0.05). (**C**) Search performance. The success rate per unit time was calculated based on the results of A and B.

detection frequency at each point of the odor field in this study is listed in the Supplementary Materials (*Figure 1—figure supplement 1*). The velocity distribution at each odor detection frequency is represented by a box plot. The red box plots in each figure represent the maximum values, and the circled dots in each box plot are the average values. The dotted line in *Figure 5A/B* is the result of

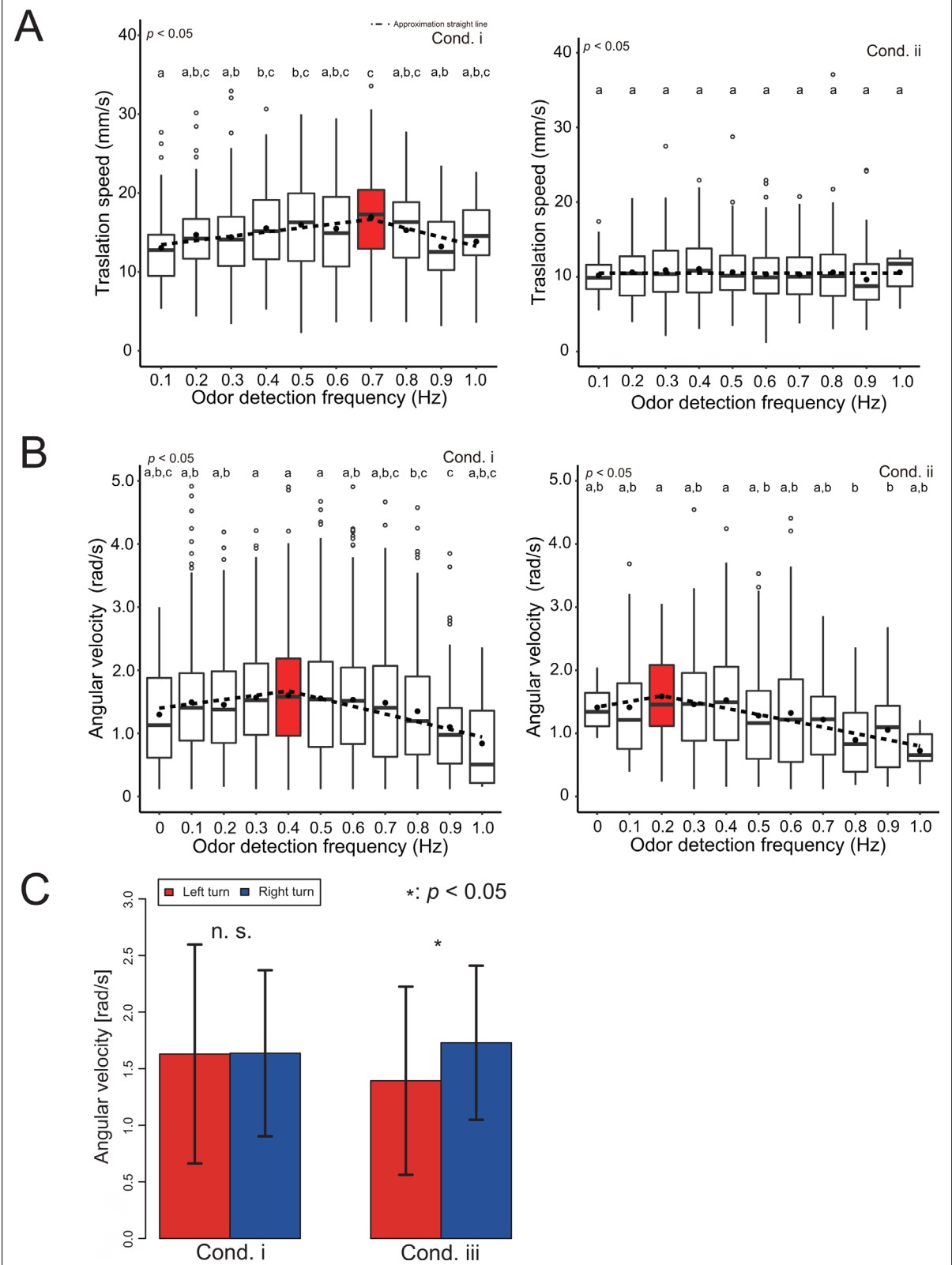

**Figure 5.** Analysis of the effects of wind and vision direction on behavior. (**A**) Changes in the translational velocity and angular velocity when the wind direction is presented correctly (cond.**i**). Both translation and angular velocity peak, and the behavior is modulated depending on the odor detection frequency. (**B**) Changes in translational velocity and angular velocity when the wind is presented from the direction opposite to the actual direction (cond. ii). The translational velocity is constant regardless of the odor detection frequency, and the peak position of the angular velocity is 0.5 times

*Figure 5 continued on next page*

*Figure 5 continued*

that of cond. i. (**C**) Comparison of angular velocities when the visual information is presented correctly (cond. **i**) and when it is presented in the opposite direction (cond. iii). Vision was used to equalize the speed of the left-right rotation.

The online version of this article includes the following figure supplement(s) for figure 5:

**Figure supplement 1.** Histogram of the angular velocity of cond. i and cond. iii.

performing a least-square approximation on the average value. The result of this least-square approximation is utilized in the mathematical modeling. According to the results of the speed change of cond. i, the translational speed increases monotonically up to the odor detection frequency of 0.7 Hz and decreases after 0.7 Hz. Moreover, the angular velocity increases monotonically up to 0.4 Hz, and then decreases. Considering that the pheromone release frequency of the silk moth female is about 0.8 Hz, when the wind is received from the same direction as the odor, the male silkmoth may actively move until it approaches the pheromone release frequency. At higher frequencies ( > 0.8 Hz), the male silkmoth reduces its movement speed to facilitate a mate search because there is a possibility that the female is nearby.

According to the results of the translational speed of cond. ii, the moth always moves at a constant speed regardless of the odor detection frequency. We also found that the frequency at which the maximum value occurs for the angular velocity was shifted to 0.2 Hz. This may be because the peak of the angular velocity shifts toward a lower odor detection frequency, and if the odor and wind direction do not match, the silkmoth rotates slowly to carefully search for the odor. This is probably because if the odor and wind direction do not match, the odor is carefully searched for by slowing down the overall movement speed.

Next, we investigated the effects of visual stimuli on odor-source search behavior. Section 2.1 suggests that visual stimuli do not directly contribute to search performance. However, many flying insects with compound eyes use visual information for their own postural control (*Dyhr and Higgins, 2010*; *Dyhr et al., 2013*). Because the silkmoth, which may retain some vestiges of flying insects (*Kanzaki, 1998*; *Shigaki et al., 2016*), also changes its rotational behavior in response to a visual stimulus presented as an optical flow (*Pansopha et al., 2014*), it is possible that the silkmoth utilizes the visual information as the input amount for postural control. To test this hypothesis, we compared the angular velocities of silkmoths in cond. i (all stimuli in the forward direction) to those in cond. iii (visual stimuli presented in the inverse direction). *Figure 5C* shows a histogram of the left and right angular velocities under each experimental condition. The red color in *Figure 5C* indicates the angular velocity when rotating counterclockwise, and the blue color shows the angular velocity when rotating clockwise. The position of the vertical bar above the histogram represents the average angular velocity, and the length of the horizontal bar represents the standard deviation. Although the average angular velocities of the left and right rotations are the same in cond. i (visual stimuli presented correctly), the angular velocities of the left and right rotations differed when the visual stimuli were presented in the opposite direction to reality (Wilcoxon rank sum statistical test, p < 0.05). These results indicate that silkmoths, like flying insects, use visual stimuli for postural control.

We found that the silkmoth modulated its behavior based on whether or not the direction of odor and wind detection coincided.

## Modeling and validation of behavioral modulation mechanisms

Here, we investigated how behavioral modulation informed by behavioral experiments using a VR system contributes to the odor-source search. A silkmoth moves by walking on a two-dimensional plane with six legs, but it does not move in a lateral direction. Therefore, we assumed that it has non-holonomic constraints and constructed our model to output straight-ahead and rotational movements. Previous studies have proposed a silkmoth search model called the surge-zigzagging algorithm (*Ryohei et al., 1992*; *Shigaki et al., 2020*), which makes action decisions using only olfactory information. In this study, we updated the surge-zigzagging algorithm to incorporate wind direction information. For convenience, we called this algorithm MiM2 (the multisensory input-based motor modulation) algorithm. A block diagram of the constructed MiM2 algorithm is presented in *Figure 6A*. The model has a straight-ahead speed (v) controller (*Equation (1)*) and an angular velocity () controller

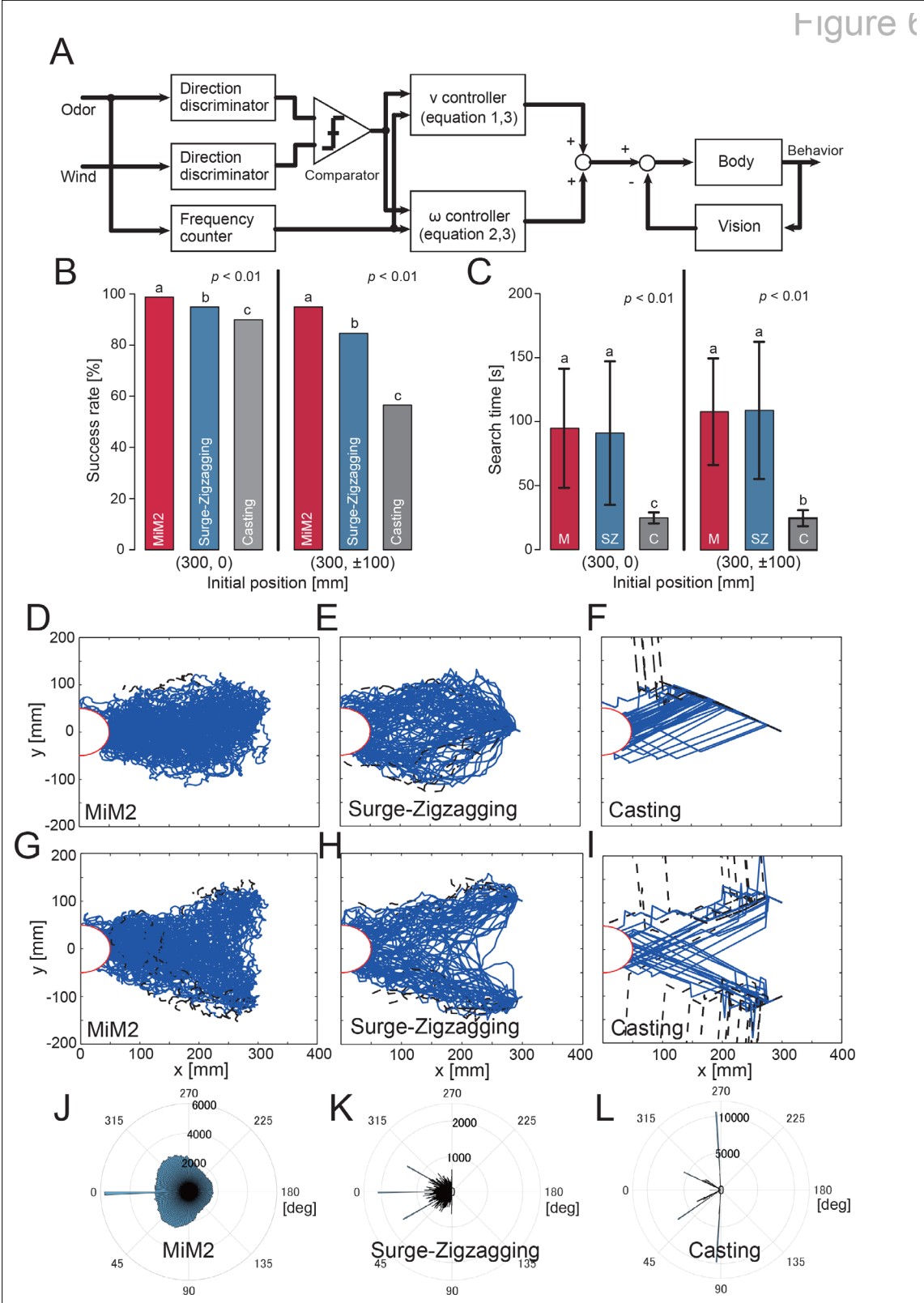

**Figure 6.** Block diagram of the proposed model and the simulation results. (**A**) Control model that modulates speed according to the degree of coincidence between odor detection direction and wind direction. (**B**) The success rate of navigation. (**C**) The search time at the time of success. (**D—I**) The result of trajectories under each experimental condition. The blue and black lines are the trajectories of successful and unsuccessful searches, respectively. (**J—L**) The result of heading angle histograms under each experimental condition. The amplitude of the heading angle histogram displayed

*Figure 6 continued on next page*

*Figure 6 continued*

in polar coordinates indicates the frequency: the higher the frequency, the more the agent moves in that direction. Note that the $0°$ direction is the windward direction and does not necessarily move toward the odor source.

The online version of this article includes the following video and figure supplement(s) for figure 6:

**Figure supplement 1.** Flowchart of other algorithms.

**Figure 6—video 1.** Experimental video of the simulation.

https://elifesciences.org/articles/72001/figures#fig6video1

**Figure supplement 2.** Simulation results when the left and right angular velocities are out of balance in rotational motion.

(*Equation 2*), and each controller changes output depending on the amount of sensory input. The detailed equations for each controller are shown below:

$$v(t) = \frac{K_v(f) \times \exp(-(t - t_d))}{1 + \exp(\gamma((t - t_d) - \beta))} \tag{1}$$

$$\omega(t) = \frac{\omega_0 \times R_d(0)}{1 + \exp(\gamma((t - t_d) - \beta))} + \frac{K_\omega(f) \times R_d(N)}{1 + \exp(-\gamma((t - t_d) - \beta))} \tag{2}$$

$$K_{v/\omega}(f) = a_{v/\omega} \times f + b_{v/\omega} \tag{3}$$

The free parameters were set to $\gamma = 1000$, $\beta = 0.50$, and $\omega_0 = 0.57$. Here, $f$ denotes the odor detection frequency. Moreover, $K(f)$ is a linear function that varies with the direction of odor and wind detection. The parameters a and b of $K(f)$ are shown in *Table 1*. These were obtained by approximating the results of the behavioral experiment using the least-squares method (dotted line in *Figure 5A/B*). In addition, $t_d$, $R_d$, and $N$ represent the odor detection timing, the odor detection direction, and the number of turn motions, respectively; $R_d(0)$ represents a straight-ahead motion state and takes a value of –1 when the left antenna detects and one when the right antenna detects; if both antennae detect, a value of –1 or one is randomly selected; $N$ increases according to *Equation (4)* up to four, but does not take a value of four or more because the number of zigzagging motions is approximately three (*Ryohei et al., 1992*). When it receives an odor stimulus according to *Equations (1–3)*, it generates a straight motion for 0.5 s and then makes a turn motion.

$$N(t) = [0.0116(t - t_d)^3 - 0.199(t - t_d)^2 + 1.1971(t - t_d) + 0.4482] \tag{4}$$

By passing through the directional discriminator and frequency counter, the odor is converted into information such as whether it was detected by the left or right antennae and how much odor it was exposed to. The wind is converted into wind direction information. A comparator is used to determine whether the direction of odor detection and the direction of the wind are the same, and the results are input to the straight-ahead speed and angular velocity controllers, which also receive odor detection frequency. Visual information controls the angular velocity of the left and right rotational movements, following which olfactory behavior takes place. From this information, the movement speed output is calculated, and the behavior is generated. We compared the MiM2 algorithm to the previous surge-zigzagging algorithm and the casting algorithm, which uses wind information for searching in a moth-inspired algorithm (*Li et al., 2016*).

The simulation environment employed a virtual environment similar to that used in the behavioral experiments in the VR system (see *Figure 6—video 1*). For each algorithm, we performed 1000 odor-source search experiments to evaluate search success rate and trajectory. In the

**Table 1.** List of parameters for $K(f)$.

(a)

| $k_v$ | | $a_v$ | $b_v$ |
|---|---|---|---|
| Odor direction =Wind direction | $f \leq 0.7$ | 5.36 | 12.9 |
| | $f > 0.7$ | –11.4 | 24.6 |
| Odor direction ≠ Wind direction | $f \leq 0.7$ | 0.0 | 10.5 |
| | $f > 0.7$ | 0.0 | 10.5 |

(b)

| $k_\omega$ | | $a_\omega$ | $b_\omega$ |
|---|---|---|---|
| Odor direction = Wind direction | $f \leq 0.4$ | 0.638 | 1.40 |
| | $f > 0.4$ | –1.22 | 2.16 |
| Odor direction ≠ Wind direction | $f \leq 0.4$ | 0.870 | 1.42 |
| | $f > 0.4$ | –0.997 | 1.80 |

simulation experiment, we set two scenarios to verify each algorithm. One was the same initial position $(x, y) = (300, 0)$ [mm] as in the biological experiment, and the other was the initial position moved $\pm 100$ mm in the crosswind direction $(x, y) = (300, \pm 100)$ [mm]. Because the latter was near the edge of the area where the odor reaches, deviation behavior possibly occur frequently. *Figure 6B and C* show the search success rate and the search time, respectively. The MiM2 algorithm showed a higher search success rate than the other algorithms, regardless of the initial position (Fisher's exact test, p < 0.01). The casting algorithm showed the shortest search time (Steel-Dwass test, p < 0.01) because it is an algorithm that actively moves upwind using odor detection. However, if the heading angle when moving upwind was incorrect, the search success rate of this algorithm was the lowest of the three, because there was a high possibility that the agent would have moved in the wrong direction. Next, we focused on the trajectory and heading angle change shown in *Figure 6D—L*. The blue and black lines in *Figure 6D—I* are the trajectories of successful and unsuccessful searches, respectively. Moreover, the amplitude of the heading angle histogram (*Figure 6J—L*; displayed in polar coordinates) indicates the frequency. For the readability of the trajectory and heading angle histogram, we randomly selected 100 trials of the 1000 replicates and plotted them. Regardless of the initial position, the search trajectory indicates that the MiM2 algorithm always places the searching agent in the middle of the odor distribution, whereas the conventional surge-zigzagging and surge-casting algorithms tend to place the searching agents near the edges of the odor. According to the heading angle histogram, the MiM2 algorithm resembles the actual silkmoth (*Figure 2*), but the other algorithms produced completely different histograms.

In order to quantitatively evaluate the trajectory of 1,000 simulations, we visualized the trajectory data (*Figure 7A—F*). The black line in *Figure 7A—F* shows the range that the odor has a high probability of reaching. Moreover, we evaluated the differences in trajectory between algorithms quantitatively by calculating their EDMs (*Figure 7H*). The comparison of these EDM values suggests that the MiM2 algorithm most closely resembles the actual silkmoth movements (*Figure 7G*) and was the closest to the results of behavioral experiments of the silkmoths while free walking. Thus, the MiM2 algorithm can simulate search behavior similar to that of real silkmoths by modulating the movement speed based on wind information in addition to odor, and equalizes the left-right angular velocities based on visual information.

## Discussion
### Virtual reality for insect behavioral measurements

All animals, including insects, need to navigate for survival and reproduction. It is especially important to navigate efficiently in harsh environments and in situations in which there are many competitors. Previous studies have investigated the homing behavior of a desert ant (*Cataglyphis*) (*Wehner, 2003*), the pheromone source localization behavior of a male silkmoth (*Bombyx mori*) (*Obara, 1979*), and the sound source localization behavior of female crickets (*Gryllus campestris L.*) (*Schmitz et al., 1982*). These studies have contributed to our understanding of the sensory-motor integration mechanism that converts sensory input into motor output – an important function of the neural system. However, the insects in these studies may not have displayed navigation behavior as they would have under natural conditions, because stimuli were controlled and presented in a fixed amount and at regular intervals in the behavioral experiments. In addition, the nature of behavioral change in a large-scale space and over a long period is relatively unknown because the behavioral experiments were carried out in a relatively small space and over a relatively short time. Because of this, multimodal virtual reality (VR) systems have attracted a great deal of attention (*Kaushik et al., 2020*; *Naik et al., 2020*). The advantage of using a multimodal VR system is that not only can behavior be measured on a large spatiotemporal scale, but the transmission of stimuli can be precisely controlled, and the behavioral output can be precisely measured. Biological data describing the relationship between sensory input and motor output are useful not only for clarifying biological functions, but also for the field of robotics, because these data play a very important role in modeling insect systems. Because the multimodal VR system in previous studies was developed for visually-dominated navigation, it was difficult to measure olfactory-dominated navigation, which was the focus of our research. Therefore, we developed a novel multimodal VR system that allowed us to measure changes in navigational behavior when the olfactory, visual, and wind directions of the silkmoth were modified. Conventionally, previous

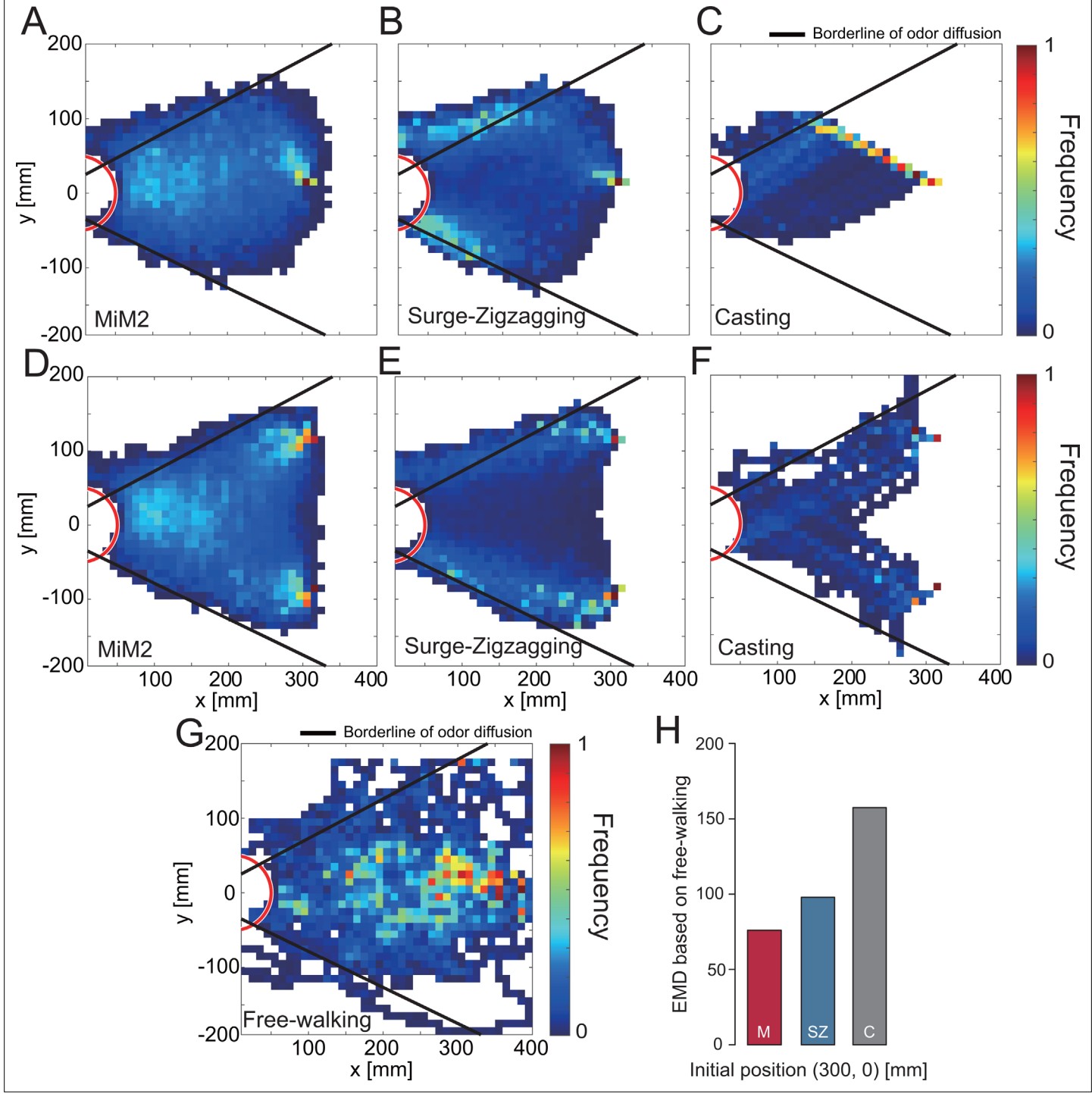

**Figure 7.** Evaluation of simulation experiments using migration probability map.
(A—F) The migration probability map at the time of success in the simulation. (G) The migration probability of an actual silkmoth when searching for a female by free walking. (H) The EMD of each search algorithm, calculated from the migration probability map from the free-walking experiment (A—C vs.G).

studies on the silkmoth have shown that (1) a 'mating dance' was elicited in response to sexual pheromones (*Obara, 1979*), (2) the condition in which visual stimuli was presented immediately after the reception of sex pheromones influenced the subsequent rotational behavior (*Pansopha et al., 2014*), and (3) behavioral inhibition occurred in response to frontal winds (*Shigaki et al., 2019*). However,

the integration of the above phenomena during navigation has not been investigated. By presenting three types of sensory information simultaneously as well as continuously, we were able to clarify the roles of each type of sensory information in navigation. In the behavioral analysis of a flying moth, it was found that when the moth detected the odor, it moved in an upwind direction (*Willis et al., 2011*; *Willis and Arbas, 1991*); and wind information is important for behavior determination. Even in the case of the silkmoth, which is a walking insect, the success rate is higher in the condition that the wind information is correctly provided (cond. 2) than in a non-correct condition (cond. 4) or in a no-wind condition (cond. 1) (*Figure 4A*). However, it does not mean that the silkmoth cannot reach the odor source when wind information is not correctly provided, and the search success rate was about the same as in the no-wind condition (approximately 60%). The wind information is therefore important, but if the wind information is unreliable, the priority of the odor information may be increased and the silkmoth performs the navigation. When the phenomena obtained by the VR experiment were mathematically modeled and the performance was confirmed by the simulation experiment, not only the original search trajectory of the silkmoth could be reproduced, but the search performance was improved, compared to the conventional algorithms.

## Behavioral modulation to odor frequency

Organisms of all size scales rely on the ability to locate an odor source in space. The chemotaxis of bacteria and a nematode have been determined by measuring and analyzing the relationship between the chemical stimulus input and behavioral output under a controlled environment (*Berg, 2008*; *Lockery, 2011*). These studies found that as the concentration of chemical stimuli increased, the probability of rotating behavior decreased linearly. In other words, bacteria or nematode chemotaxis followed an odor gradient, which allowed them to locate the odor source. In the space where bacteria and the nematode exist, chemotaxis is effective in part because the odor field of the environment does not change significantly due to wind. In the case of larger size animals, the odor field of the environment is quite complex and it is difficult to reach the odor source by following the gradient alone. Odor fields can be complex because odor molecules are transported by airflow and mixed with other molecules at their destinations, forming complex structures (*Crimaldi and Koseff, 2001*; *Murlis et al., 1992*). In addition, the odor molecules themselves are discrete in space, and do not carry information about the source of the odor. However, a study using a sensor array to measure the arrival of odors carried by the wind revealed that the odors emitted from the source arrive periodically (*Kikas et al., 2001*; *Murlis et al., 2000*). Moreover, the periodicity is correlated to some extent with the distance from the source; the closer the source, the shorter the cycle, and the farther the source, the longer the cycle of arrival. Neither the concentration of the odor nor the number of exposures played an important role in the search process in *Drosophila*, but the 'tempo' at which flies encountered an odor was an important factor in the decision-making process (*Celani, 2020*; *Demir et al., 2020*). If we assume that the 'tempo' is the rate of odor detection per unit time, it is related to the cycle in which the odor arrives. We hypothesized that the silkmoth, like flying insects such as *Drosophila*, modulates its behavior based on the 'tempo' of the odor, therefore we included odor frequency in our analyses.

When the direction of the wind and the odor coincided, movement speed peaked when the odor frequency was 0.7—0.8, which is similar to the frequency at which a female silkmoth releases sex pheromones (0.79 ± 0.05 Hz) (*Fujiwara et al., 2014*). Based on these findings, we hypothesized that if the male silkmoth detects wind and odor from the same direction, it correctly moves in the direction of the female and actively searches the field until it reaches a location where the sex pheromone release frequency approximates that of the female. When the frequency exceeds 0.8 Hz, the male seems to be in the vicinity of the female and therefore increases the spatiotemporal resolution of the search in order to locate the female and prepare for the transition to mating behavior. This might correspond to the silkmoth switching to the odor source declaration algorithm in olfactory navigation. The in vivo data obtained in the current study are consistent with those of earlier research demonstrating that an algorithm that shortens the travel distance of the surge toward the end of the search improves the search performance of the robot (*Shigaki et al., 2018*).

In an earlier study investigating the direction of wind and odor in the environment, the direction of wind and odor were the same in the open-field experiment (no obstacles) (*Murlis et al., 2000*). However, the direction of wind and odor is not always the same in a complex environment, such as a forest with many trees (*Murlis et al., 2000*). Our data showed that in situations where the wind and

odor direction do not match, the silkmoth always moves at the same rate, given any odor detection frequency. This may be a chemical tracking strategy to avoid leaving the odor range by moving at a speed lower than normal speed and suggests that the silkmoth is estimating the degree of environmental turbulence and modulating its behavior based on the odor and wind direction information. This strategy may be applicable to an engineering search system that switches the search strategy according to the environmental conditions.

## Comparison with conventional CPT algorithm

Odor sensing plays an important role in situations in which a visual search is difficult (e.g. a space filled with darkness and thick smoke). However, because the technology for the development of an odor sensor is lagging behind that of other sensory sensors (e.g. cameras, microphones), robot olfactory research is still a developing field. Currently, dogs play the role of odor sensors, but due to the high cost of training dogs and the deterioration of their odor sense with physical condition and age, engineering solutions using autonomous mobile robots for searching are needed. Conventional robot olfactory research has emphasized the development of motion planning (a search algorithm). The search algorithm is roughly divided into two fields: one is a bio-inspired algorithm that imitates the search behavior of living organisms, and the other is a search that estimates the position of the odor source using a statistical method (statistical algorithm). The bio-inspired algorithm, which mimics the behavior of organisms that can search in real time, is superior to the statistical algorithm in terms of robot implementation (*Russell et al., 2003*; *Lochmatter et al., 2008*). However, behavior patterns and movement speeds in these bio-inspired algorithms are always constant, regardless of the environmental conditions. Accordingly, there is a problem that the original search performance of living things cannot be replicated. For this reason, different kinds of research have been carried out more recently and applied to the bio-inspired algorithm, such as information-theoretic analysis of the trajectory of an insect (*Hernandez-Reyes et al., 2021*), extraction of adaptability from neuroethology data by fuzzy inference (*Shigaki et al., 2020*), and the acquisition of behavioral switching indices in response to environmental changes by measuring insect behavior while visualizing airflow (*Demir et al., 2020*). In the current study, we found that behavioral modulation occurred based on the relationship between the direction of the odor arrival and the wind, and we used our data to reproduce a behavioral trajectory that was not only better than that of the conventional bio-inspired algorithm but was also more similar to the search behavior of the actual silkmoth. Therefore, it is clear that a probabilistic and time-varying behavior modulation mechanism has a better search performance than a time-invariant search algorithm.

# Materials and methods

## Animals

Silkmoths (*Bombyx mori*; Lepidoptera: Bombycidae) were purchased from Ehine Sansyu Co., Japan. Adult male moths were cooled to 16°C one day after eclosion to reduce their activity and were tested within 2-7 d of eclosion. Before the experiments, the moths were kept at room temperature (25-28°C) for at least 10 min.

## Virtual odor field

To generate a virtual odor field closely mimicking reality, smoke was emitted into the actual environment. The diffusion of the smoke in a two-dimensional plane was recorded and implemented into the simulator using image processing. The smoke visualization experiment was conducted in an approximately 2.5 × 0.8 m area in a darkroom. First, the smoke was visualized using a smoke generator (particle diameter: < 10 μm) and a laser sheet (1 mm in thickness) (*Figure 1—figure supplement 1A*). The smoke emitted into the room gleamed when it hit the irradiated laser sheet, and the distribution of smoke was recorded using a high-sensitivity camera. Note that the behavior of smoke reflected by the laser could be observed in two dimensions because the laser was in the form of a sheet. In this experiment, we carried out a visualization experiment in a darkroom because it is important to accurately record the light reflected by the smoke. This method was based on particle image velocimetry (PIV), which measures the velocity of fluid flow in space by visualizing particles and analyzing their movements. PIV accuracy is improved by scattering a large amount of smoke (particles) in space

because it focuses on accurately tracking each particle; however, our method did not require following each particle. Instead, our method used a mass of smoke as an odor plume, and we observed how smoke floated and was distributed in space. In the visualization experiment, the flow meter (1.0 L/min) and solenoid valve (1 Hz) were controlled under the same release conditions as in the behavioral experiment. Our method required less smoke than PIV, and the smoke was thinner and more difficult to visualize. For this reason, it was necessary to increase the sensitivity of the camera; however, noise increased accordingly. In PIV, it is not necessary to distinguish between smoke particles and dust because only the movement of particles in space is important, but our method required removing other particles in order to measure only the smoke position as the plume. Hence, we applied image processing to the smoke image. Luminance values can have a large range because the intensity of smoke varies with airflow and time. In addition, it is difficult to extract only smoke with simple thresholding because the noise caused by dust has the same luminance as does smoke. Therefore, we adopted a method that focuses on connected components to remove noise while maintaining the shape of the smoke. In groupings based on connected components, two objects were considered to be connected if adjacent pixels in a binary image take a value of 1. Because smoke exists as a mass, it can be inferred that it has an area above a certain level when divided into connected components. For this reason, we removed the pixels whose connected components were not in adjacent pixels and whose area did not exceed a certain level. These processes were performed with source code using OpenCV.

## Configuration of virtual reality system

The behavioral experiment was carried out using a homemade virtual reality system (a photograph of the actual device is shown in *Figure 1A*). The traditional tethered measurement system was used for behavior measurement, and a stimulator that presented odor, visual, and wind direction information was installed around the tethered measurement system (*Figure 1—figure supplement 1A*). The details of each stimulator were as follows:

### Odor stimulator

We provided sex pheromone stimulation to both antennae of a silkmoth using Bombykol ((E,Z)–10,12-hexadecadien-1-ol). A cartridge containing 1000 ng of bombykol was placed in a tube in order to present air containing bombykol to the silkmoth. The compressed air from the air compressor (NIP30L, Nihon Denko, Aichi, Japan) passed through three gas washing bottles containing degreased cotton, activated carbon, and water, respectively, and was adjusted to 1.0 L/min using a flow meter (KZ-7002–05 A, AS ONE CORPORATION, Osaka, Japan). The timing of the stimulation was controlled by switching the flow path of the solenoid valve (VT307, SMC Corporation, Tokyo, Japan). The odor stimulus discharge ports were integrated with the tethered rod, and the discharge ports were located above the antennae. The tethered rod with the discharge ports was fabricated using a 3D printer (Guider2, Flashforge 3D Technology Co., Ltd., Zhejiang, China). As a result of presenting a one-shot pheromone stimulus to the silkmoth using this odor stimulator, the odor stimulus was presented correctly because female search behavior was elicited (*Figure 1—figure supplement 1B*).

### Vision stimulator

The VR system in a previous study focused on navigation behaviors including object recognition, so that they provided visual stimuli using a monitor (*Naik et al., 2020*; *Radvansky and Dombeck, 2018*; *Kaushik et al., 2020*). However, in this study, we provided the direction in which the landscape was flowing as a visual stimulus, not object recognition using vision. For this reason, we employed an optical flow as a visual stimulus for our VR system. Because the optical flow presentation method using LED arrays was also used in previous studies with flying insects (moth and bee), we used the same method in this study (; *Zheng et al., 2019*). The visual stimulus device was constructed by arranging LEDs (WS2812B, WORLDSEMI CO., LIMITED, GuangDong, China) with a built-in microcomputer in an array around the tethered measurement system. The LED array was comprised of 256 LEDs arranged in 8 (vertical) × 32 (horizontal) panels. The array was presented as an optical flow by controlling the lighting timing of each LED according to the angular velocity of the silkmoth.

Because it has been reported that the silkmoth causes optomotor reflexes with respect to optical flow and its neck tilts in the direction of optical flow (*Minegishi et al., 2012*), whether or not this vision

stimulator was functioning was evaluated by the angle of the neck with respect to optical flow. The vision stimulator was functioning properly because the silkmoth has the largest neck tilt in the range of angular velocity during female search behavior (*Figure 1—figure supplement 1C*), and this result is similar to the past research data (*Pansopha et al., 2014*).

## Wind stimulator

The wind presented to the silkmoths was generated using a push-pull rectifier (*González et al., 2008*). A push-pull rectifier consists of a push side that sends out the wind and a pull side that draws in the sent wind. Fans (PMD1204PQB1, SUNON, Takao, Taiwan) were installed on both the push and pull sides to generate airflow. Space can be used effectively using this device because air can flow without covering the workspace with a partition. The push-pull rectifier was connected to a hollow motor (DGM85R-AZAK, ORIENTAL MOTOR Co., Ltd., Tokyo, Japan) which rotated to generate wind. The performance of the push-pull rectifier was evaluated using particle image velocimetry (PIV) (*Adrian, 2005*). For PIV analysis, we used videos that were shot at 800 fps with a resolution of 640 × 480 pixels. As a result, it was confirmed that the wind generated by the push-pull rectifier did not generate vortices (*Figure 1—figure supplement 1D*).

## VR system evaluation experiment

We experimentally verified the extent to which the VR system equipped with odor, vision, and wind direction stimulators could reproduce a free-walking experiment in a real environment. In the free-walking experiment in a real environment, the search field was the same size as the VR system, and an odor emission frequency (1 Hz) was used. Two replicated experiments were carried out using 15 moths, and their behavior was recorded using a 30 fps video camera (BSW200MBK, BUFFALO, Aichi, Japan). The figure shows the quantitative comparison of search success rate and relative distance (*Figure 1—figure supplement 1E*, F). We found that there was no difference between the results of VR and the free walking experiment. In other words, VR was able to reproduce the experiment of free walking, and the behavioral experiment could be performed using the VR device proposed in this study.

## Migration pathway ratio map

A migration pathway ratio map was used to statistically process the trajectories of all trials and denoted the points where the quadcopter frequently passed through the experimental field. We created the migration pathway ratio map $C_n(x, y)$ based on the rule given by *Equation (5)*. Here, $n$ represents the trial number, and the size of the grid cell was 0.1 × 0.1 cm.

$$C_n(x, y) = \begin{cases} 1 : \text{The silkmoth passed through the coordinates (x, y).} \\ 0 : \text{The silkmoth did not pass through the coordinates (x, y).} \end{cases} \quad (5)$$

We applied the equation to all trial trajectories and summarized them, and then calculated the trial average based on *Equation (6)* to create the migration pathway ratio map $P(x, y)$. The migration pathway ratio map was created using MATLAB (2020a, MathWorks, MA, USA).

$$P(x, y) = \frac{\sum_{n=1}^{N} C_n(x, y)}{N} \quad (6)$$

## Statistical analysis

For all data analyses, R version 4.0.3 was used (R Core Team). All earth mover's distance calculations were performed using the Python 3.7 language.

## Odor-source search algorithm

The details of the three algorithms for which simulation experiments were performed are described here:

## Surge-zigzagging

A surge-zigzagging algorithm models the female search behavior of an adult male silkmoth, which is a walking insect (*Figure 6—figure supplement 1*). In response to a one-shot sex pheromone stimulus, the silkmoth exhibits a movement that advances in the direction of the odor and a straight movement (surge), followed by a rotational movement (zigzag/loop) to search for further odor information from all directions. The flow chart of the surge-zigzagging algorithm is shown in the supplementary figures. The surge state in the algorithm lasts for 0.5 s because surge behavior lasts for about 0.5 s after the odor stimulus is presented. Because the surge behavior is elicited when the odor stimulus is presented, the trajectory becomes linear when the odor is continuously (high frequency) presented.

## Casting

A casting algorithm models the odor source localization behavior of a flying moth by mapping it onto a two-dimensional plane (*Li et al., 2016*). A schematic diagram of this algorithm and an implemented flowchart are shown in *Figure 6—figure supplement 1*. When the agent in this algorithm detects an odor plume, it moves upwind, and when it loses sight of the plume, moves in the crosswind direction to rediscover the plume. At this time, when moving upwind, it moves at a certain angle $\beta$ with respect to the upwind direction and continues moving a certain distance $d_{lost}$ after losing sight of the plume. By repeating these movements in the upwind and crosswind directions, the odor source is localized. The parameters of $\beta$ and $d_{lost}$ are 30° and 2.5 cm, respectively, and these coincide with the highest search success rates in previous research (*Lochmatter et al., 2008*).

## Acknowledgements

We thank Dr. Takeshi Sakurai (Tokyo University of Agriculture) for providing the sex pheromone, bombykol.

## Additional information

### Funding

| Funder | Grant reference number | Author |
|---|---|---|
| Japan Society for the Promotion of Science | JP19H04930 | Shunsuke Shigaki |
| Japan Society for the Promotion of Science | JP19K14943 | Shunsuke Shigaki |
| Japan Society for the Promotion of Science | JP19H02104 | Daisuke Kurabayashi |

The funders had no role in study design, data collection and interpretation, or the decision to submit the work for publication.

### Author contributions

Mayu Yamada, Data curation, Formal analysis, Software, Validation, Visualization, Writing – original draft; Hirono Ohashi, Data curation, Formal analysis, Methodology, Visualization, Writing – review and editing; Koh Hosoda, Investigation, Methodology, Supervision, Writing – review and editing; Daisuke Kurabayashi, Conceptualization, Funding acquisition, Investigation, Resources, Validation, Visualization, Writing – review and editing; Shunsuke Shigaki, Conceptualization, Formal analysis, Funding acquisition, Investigation, Methodology, Project administration, Resources, Software, Validation, Visualization, Writing – original draft, Writing – review and editing

### Author ORCIDs

Mayu Yamada  http://orcid.org/0000-0002-9855-9157
Hirono Ohashi  http://orcid.org/0000-0001-5303-7200
Koh Hosoda  http://orcid.org/0000-0001-8392-1021
Daisuke Kurabayashi  http://orcid.org/0000-0002-3186-6531
Shunsuke Shigaki  http://orcid.org/0000-0002-5689-1338

## Ethics

The silkmoths experiments in this study were examined and approved by the Osaka University and Tokyo Institute of Technology Gene Recombination Experiments Safety Management Committee.

## Decision letter and Author response

Decision letter https://doi.org/10.7554/eLife.72001.sa1
Author response https://doi.org/10.7554/eLife.72001.sa2

# Additional files

## Supplementary files
• Transparent reporting form

## Data availability

All experimental data are available on Zenodo via https://doi.org/10.5281/zenodo.5724790.

The following dataset was generated:

| Author(s) | Year | Dataset title | Dataset URL | Database and Identifier |
| --- | --- | --- | --- | --- |
| Shigaki S | 2021 | Virtual reality system data of silkmoths | https://doi.org/10.5281/zenodo.5724790 | Zenodo, 10.5281/zenodo.5724790 |

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
