## [Editor Report]

This paper uses a multi-model virtual reality system to assess which combinations of visual, wind, and olfactory information male silk moths rely on to find a female. The overall conclusion is that for the moths to search effectively, wind direction information is an important input. Vision, on the other hand, while it is used to control angular velocity, does not appear to be important for the moths to search effectively. This paper is of interest to neuroscientists and engineers interested in how multimodal sensory input controls navigational behavior. The experiments and modeling effort provide an advance in our understanding of how odor and wind information is combined in male silkmoths as they search for females.

---

## [Decision Letter]

**Decision letter after peer review:**

Thank you for submitting your article "Analysis of Multisensory-Motor Integration in Olfactory Navigation of Silkmoth, *Bombyx mori*, using Virtual Reality System" for consideration by *eLife*. Your article has been reviewed by 2 peer reviewers, and the evaluation has been overseen by a Reviewing Editor and Aleksandra Walczak as the Senior Editor. The reviewers have opted to remain anonymous.

Essential revisions:

1) For Figure 1, was the smoke plume measured in 3D? If not, then the 2-dimensional projection of the 3-dimensional plume will create a substantially more continuous (less intermittent) plume than an insect would experience when moving through a 3-dimensional turbulent plume. This substantially limits the realistic nature of the experiments, and therefore the interpretation of the results. In particular, this limits how the frequency of odor detection results can be interpreted since the underlying plume model is to simplified. The idea of the smoke plume measurement to provide a realistic plume model is great, but the analysis of the plume must be done in 3D to provide a realistic experience to the moths.

2) For Figure 2, it is hard to interpret what the EMD result actually means beyond being similar or different. While this offers a nice summary measurement, to really understand what is going on, a more mechanistic analysis of the individual trajectories is needed. For example, please show individual trajectories of the moths, and make an effort to analyze their turn by turn decisions. Given that a model is presented at the end of the paper, there is a substantial opportunity to provide deeper insight by comparing the individual trajectories with the trajectories generated by the model. Specifically, it would good to see not just a better explanation of what the EMD metric is/means. The EMD metric seems to analyze what results from turn-by turn-decisions, but it would be nice to understand what those turn-by-turn decisions actually are (since there is data for that kind of analysis).

3) There is some mismatch between the model and the experiments. The model discusses relative timing of odor detections on the left and right antenna versus the direction of the wind. However, in the experiments, it is not clear whether odor stimuli were presented to have different timings for the left and right antenna. Furthermore, if this was included the in experiments, the results do not seem to be analyzed in a clear manner. A more comprehensive and clear analysis of how odor arrival time on the left and right antenna versus the supplied wind direction information influenced mechanistic behavioral decisions is needed.

4) There are huge swaths of references missing that should be considered to provide better context. Many of these references are for other insect species, namely the fruit fly *Drosophila melanogaster*, for which a great deal is known about both olfactory search, wind sensing, and multi-sensory integration (vision + olfaction, in particular, but also mechano-visual integration). Here are some PI's the authors should read up on, and choose several references from each to include: Rachel Wilson, Kathy Nagel, Mark Willis, Mark Frye, Michael Dickinson, Tom Daniel. A useful review to help collect some of these references: Algorithms for Olfactory Search across Species, Baker et al., 2018, though there are several papers relevant to this study that have been published since this review as well. In addition, the use of VR for exploring insect behaviors like search and navigation is hardly a new development at this point in time. Indeed, work on flies, *Drosophila* and otherwise, going back decades has used VR to extract principles regarding issues such as visually mediated flight control or walking. More pertinent to the ideas of multimodal sensory integration explored here, over the past decade numerous researchers have combined visual input with odor and other cues to discern the relative importance of each of these modalities during search behavior. For example, see Duistermars and Frye, 2008. Generally, the reviewers felt that the paper overemphasizes the technical advance without providing sufficient biological context. So much work has been done on Bombyx that a paper using these methods has the ability to address, but much of that literature is absent from the paper. Focusing more on the behavior will broaden the appeal of the paper by putting it in conversation with a well-established phenomenon.

5) While the model is well-done and fits with the goals of the paper. Asking about the role of wind direction in this behavior is an important step given the behavioral data presented. However, the reviewers were not convinced, based on the presented data, that the new model developed by the authors is much better than the surge-zigzag model. The success rates are slightly different (is the statistical difference a function of the number of runs or timesteps?) and both models search about the same amount of time before finding the source. Finally, the migration probability maps are rather similar, so it is hard to conclude that factoring in wind direction is necessary to get good performance out of the model.

6) Overall, the manuscript would greatly benefit from professional editing to improve the writing. Many sections read more as stream of consciousness than tightly edited scientific prose. Some minor writing-related issues are outlined in the comments below, but this list is far from exhaustive. Improved writing would greatly advance the scientific impact of the work.

---

## [Author Response]

Essential revisions:1) For Figure 1, was the smoke plume measured in 3D? If not, then the 2-dimensional projection of the 3-dimensional plume will create a substantially more continuous (less intermittent) plume than an insect would experience when moving through a 3-dimensional turbulent plume. This substantially limits the realistic nature of the experiments, and therefore the interpretation of the results. In particular, this limits how the frequency of odor detection results can be interpreted since the underlying plume model is to simplified. The idea of the smoke plume measurement to provide a realistic plume model is great, but the analysis of the plume must be done in 3D to provide a realistic experience to the moths.

Thank you for your valuable comments. We apologize for the inadequate explanation. We captured the smoke plume movement on a two-dimensional plane as described in the virtual odor field of the “Materials and methods” chapter (section 4.2). We set a laser sheet with a thickness of about 1 mm at a distance of 10 mm from the ground. This height of the laser was approximately equal to the height of the silkmoth antennae from the ground. The smoke particles (< 10 µm) were emitted into the space at a cycle of 1 Hz. The movement of the smoke at the height of the antennae of the silkmoth was reproduced in the virtual environment because only the smoke reflected on this laser sheet was photographed by the high-sensitivity camera. Note that the movement of the silkmoth’s head was just a few millimeters at the maximum; therefore, this was sufficient to reproduce the behavior of smoke reflected on a two-dimensional plane in this paper. Of course, if we want to use the same virtual odor field for flying insects, we can reproduce the three-dimensional realistic smoke behavior by preparing multiple laser sheets and measuring them. We added these explanations and an outline of the experimental setup of capturing the smoke.

Revised part: “Virtual odor field”, Figure 1—figure supplement 2

2) For Figure 2, it is hard to interpret what the EMD result actually means beyond being similar or different. While this offers a nice summary measurement, to really understand what is going on, a more mechanistic analysis of the individual trajectories is needed. For example, please show individual trajectories of the moths, and make an effort to analyze their turn by turn decisions. Given that a model is presented at the end of the paper, there is a substantial opportunity to provide deeper insight by comparing the individual trajectories with the trajectories generated by the model. Specifically, it would good to see not just a better explanation of what the EMD metric is/means. The EMD metric seems to analyze what results from turn-by turn-decisions, but it would be nice to understand what those turn-by-turn decisions actually are (since there is data for that kind of analysis).

Thank you for your helpful comments. As you pointed out, we added trajectory figures of all the trials. Moreover, the heading angle change during navigation is represented by a histogram of polar coordinates. If the frequency of the heading angle histogram is high, it means that a silkmoth moves in that direction. Please note that the 0° direction is the + x direction (upwind direction), not necessarily the direction of the odor source. Because the silkmoth searches in all directions using a turning motion, a certain frequency occurs in all directions. From trajectories and heading angle histograms, the odor-only trajectory (cond. 1) demonstrated that although the frequency of moving with the heading angle toward the windward direction was high as a whole, leaving behavior from the range with a high probability of odor arrival occurred, so that the search failed. By providing wind (cond. 2) and visual (cond. 3) stimuli in addition to odor, the deviated behaviors from the range with a high probability of odor arrival were extremely reduced; therefore, the silkmoth modulates the odor source search behavior using other sensory information.

When the wind stimuli were input, the behavior modulation applied so that the heading angle changed in the windward direction even if there was deviated behavior from an area with a high probability of odor arrival. In the case of the visual input, although there was no trajectory that deviated from the odor reach area, there were many angles with a frequency other than 0° in the histogram of the heading angles, which suggests that an omnidirectional search is conducted more carefully using posture control. As a result, we found that the search success rate of cond. 3 was lower than that of cond. 2. The heading angle histogram of cond. 4 (presented from the opposite direction to that actually received by the wind) indicates that a peak appears in the downwind direction, so that wind information has an effect on search behavior. Even when the visual presentation direction was opposite, an extreme peak in the upwind direction did not appear, and the result was that a peak also occurred in the crosswind direction. When the visual information is presented in the opposite direction (the visual information is input as if it rotates in the opposite direction), it gives the illusion that it does not rotate correctly, and it performs more rotational actions. However, because the odor has a higher priority sensory information, the search success rate does not decrease. From this, it is presumed that odor is of course an important sensory input, but information on wind direction is also important and visual information is used only for body control.

We plotted the trajectories and heading angle histograms of each trial simulated under the same initial condition as the biological experiment. As a result, we found that the search algorithm had a different trajectory from the actual moth. On the other hand, the proposed algorithm (MiM2) was found to be quite similar in trajectory and heading angle histogram. From this, we concluded that the proposed algorithm reproduced the female localization behavior of the silkmoth compared to the conventional algorithm.

Some tendency appeared in behavioral changes depending on the environmental conditions presented, but it is difficult to evaluate the overlapping parts of trajectories simply by plotting each trajectory. Hence, we visualized trajectories as a two-dimensional histogram in order to comprehensively consider which route was selected. Because a two-dimensional histogram is a kind of probability distribution, we decided to use an earth mover’s distance (EMD) that expresses the similarity between probability distributions by distance in order to quantitatively determine whether the probability of selecting the path differs or is similar depending on the environmental conditions. If the distance of the two-dimensional histogram of the trajectory is very short (EMD value is small), the same route is selected and moved, and if the distance is long, the route is selected differently. In other words, we can quantitatively evaluate the change in behavioral choices depending on the environmental conditions presented. Conventionally, it is determined whether or not the trajectories are qualitatively similar, but now we can evaluate trajectories quantitatively based on the EMD value.

Revised part: Figure 2 and 6, Line 140—162 and 293—302.

3) There is some mismatch between the model and the experiments. The model discusses relative timing of odor detections on the left and right antenna versus the direction of the wind. However, in the experiments, it is not clear whether odor stimuli were presented to have different timings for the left and right antenna. Furthermore, if this was included the in experiments, the results do not seem to be analyzed in a clear manner. A more comprehensive and clear analysis of how odor arrival time on the left and right antenna versus the supplied wind direction information influenced mechanistic behavioral decisions is needed.

We appreciate the Reviewer’s comment on this point. We utilized two solenoid valves to provide odor stimuli independently to the left and right antennae of a silkmoth. However, as you pointed out, it is not clear whether the silkmoth actually received the odor stimuli independently. Accordingly, we carried out additional behavioral experiments that eliminated the left and right odor stimulus inputs. In other words, we conducted the experiment to provide odor stimulus to both antennae of the actual silkmoth, no matter what odor stimulus was received in the virtual environment. From the results of previous behavioral experiments, the silkmoth utilized the bilateral information of odor to elicit effective female search behavior because the initial rotation direction of the zigzagging motion was determined by the timing of presentation to the left and right antennae. Therefore, we expected that the search behavior would change by eliminating this left and right odor information. We added the experimental results into the supplementary materials. By eliminating bilateral information of odor, the search success rate clearly decreased and the search time increased. From this, we concluded that the VR correctly provided the bilateral odor information and that the silkmoth executed the female search by preferentially using bilateral information. These experimental data and explanations have been added to the manuscript.

Revised part: Figure 2—figure supplement 1

4) There are huge swaths of references missing that should be considered to provide better context. Many of these references are for other insect species, namely the fruit fly *Drosophila melanogaster*, for which a great deal is known about both olfactory search, wind sensing, and multi-sensory integration (vision + olfaction, in particular, but also mechano-visual integration). Here are some PI’s the authors should read up on, and choose several references from each to include: Rachel Wilson, Kathy Nagel, Mark Willis, Mark Frye, Michael Dickinson, Tom Daniel. A useful review to help collect some of these references: Algorithms for Olfactory Search across Species, Baker et al., 2018, though there are several papers relevant to this study that have been published since this review as well. In addition, the use of VR for exploring insect behaviors like search and navigation is hardly a new development at this point in time. Indeed, work on flies, Drosophila and otherwise, going back decades has used VR to extract principles regarding issues such as visually mediated flight control or walking. More pertinent to the ideas of multimodal sensory integration explored here, over the past decade numerous researchers have combined visual input with odor and other cues to discern the relative importance of each of these modalities during search behavior. For example, see Duistermars and Frye, 2008. Generally, the reviewers felt that the paper overemphasizes the technical advance without providing sufficient biological context. So much work has been done on Bombyx that a paper using these methods has the ability to address, but much of that literature is absent from the paper. Focusing more on the behavior will broaden the appeal of the paper by putting it in conversation with a well-established phenomenon.

We thank the Reviewer for this comment. We added some papers including the one that you presented. It has been reported that fruit flies and flying moths move in an upwind direction when they encounter an odor. However, silkmoths do not necessarily move upwind, but move in the direction of odor detection. In the case of the silkmoth, the female search behavior is modulated by information on wind direction and this behavior may be equivalent to the upwind movement of fruit flies or flying moths during odor detection. The results of the current study and the above results lead us to conclude that our VR system contributed to a certain extent because we were able to deal with the problem of "it is often difficult to measure behavioral responses to controlled turbulent stimuli" in the literature.

Regarding VR research [1], the recent definition of VR is that "virtual reality creates a physical and a mental space for people." In other words, we need to connect a device that can provide multisensory stimuli to organisms (physical space) and a mental space where avatars reflect their actions. Based on this definition, we claim that most biological experiments with insects have been conducted in a physical space (just utilizing multisensory stimulators) and have not measured the purposeful behaviors that result from connection to the mental space. Behavioral experiments using a multi-sensory stimulator have revealed how each modality is utilized, but it has not been directly investigated how other modalities affect actual navigation that predominantly utilizes an odor (as in the current study). In fact, the previously proposed insect-inspired algorithm for navigation was modeled based on experimental data using a multisensory stimulus device, but as a result of simulation, it behaves far differently to the actual insect. On the other hand, the model derived from the experimental results using the VR system were able to reproduce the behavior of the insect (silkmoth). In other words, we claim that in order to model the superior functions of an organism, it is necessary to do more than provide multisensory stimuli, and it is important to obtain the relationship between multisensory stimuli and behavioral output when the organism performs the actual purposeful behavior (e.g., navigation behavior).

[1] Zhou NN, Deng YL. Virtual reality: A state-of-the-art survey. International Journal of Automation and Computing. 672 2009; 6(4):319–325.

Revised part: “Introduction”

5) While the model is well-done and fits with the goals of the paper. Asking about the role of wind direction in this behavior is an important step given the behavioral data presented. However, the reviewers were not convinced, based on the presented data, that the new model developed by the authors is much better than the surge-zigzag model. The success rates are slightly different (is the statistical difference a function of the number of runs or timesteps?) and both models search about the same amount of time before finding the source. Finally, the migration probability maps are rather similar, so it is hard to conclude that factoring in wind direction is necessary to get good performance out of the model.

Thank you for your valuable comments. Because a surge-zigzagging algorithm utilizes only an odor stimulus to determine actions, it would seem that an agent will move to the edge where the change in odor information occurs most. The behavior of MiM2 is modulated by information on wind direction, and it moves to the areas with a high probability of odor reach. Hence, we thought that the effectiveness of the model could be shown by starting the search for the agent from near the edge. Consequently, we set the initial position (x, y) = (300, ± 100) in the area where the odor is hard to reach and carried out the simulation. As a result, the MiM2 algorithm, which actively moves in the area where the probability of odor reach is high, was significantly better than other conventional algorithms. As for the search time, because MiM2 was based on surge-zigzagging, the movement speeds were almost the same; therefore, both algorithms (MiM2 and surge-zigzagging) required almost the same search time. However, we found that there was a difference in the search success rate because there was a difference in the selected route.

Revised part: Lines 282—311 at “Modeling and validation of behavioral modulation mechanisms”

6) Overall, the manuscript would greatly benefit from professional editing to improve the writing. Many sections read more as stream of consciousness than tightly edited scientific prose. Some minor writing-related issues are outlined in the comments below, but this list is far from exhaustive. Improved writing would greatly advance the scientific impact of the work.

Thank you for your helpful suggestions. We received English proofreading from an expert to improve the manuscript (Editage).